# Robot-Assisted versus Trans-Umbilical Multiport Laparoscopic Ureteral Reimplantation for Pediatric Benign Distal Ureteral Stricture: Mid-Term Results at a Single Center

**DOI:** 10.3390/jcm11216229

**Published:** 2022-10-22

**Authors:** Yanhua Guo, Yang Cheng, Dan Li, Hong Mei, Huan Jin, Guo Chen, Anpei Hu, Qilan Li, Xinyi Du, Liduan Zheng, Qiangsong Tong

**Affiliations:** 1Department of Pediatric Pathology, Union Hospital, Tongji Medical College, Huazhong University of Science and Technology, 1277 Jiefang Avenue, Wuhan 430022, China; 2Department of Pathology, Union Hospital, Tongji Medical College, Huazhong University of Science and Technology, 1277 Jiefang Avenue, Wuhan 430022, China; 3Clinical Center of Human Genomic Research, Union Hospital, Tongji Medical College, Huazhong University of Science and Technology, 1277 Jiefang Avenue, Wuhan 430022, China

**Keywords:** robot-assisted laparoscopic ureteral reimplantation, trans-umbilical multiport laparoscopic ureteral reimplantation, benign distal ureteral stricture, pediatric patients

## Abstract

Objective: Robot-assisted laparoscopic ureteral reimplantation (RALUR) and trans-umbilical multiport laparoscopic ureteral reimplantation (TMLUR) are both minimally invasive procedures for benign distal ureteral stricture (DUS). However, TMLUR has rarely been reported in published research, thus the difference in mid-term outcome of these two procedures warrants investigation. Methods: Patients who underwent RALUR or TMLUR for pediatric DUS from April 2017 to November 2020 at our institution were retrospectively analyzed and 56 patients were included in this retrospective comparison. Demographic characteristics, perioperative data and follow-up results were collected and analyzed in RALUR and TALUR groups. Results: RALUR and TMLUR were successfully performed in children aged from 12.0 to 142.0 months, without conversion to open ureteral reimplantation. RALUR took shorter operative time than TMLUR (*p* = 0.005) with less blood loss (*p* = 0.001). Meanwhile, patients receiving RALUR encountered a greater financial burden (*p* < 0.001) with less cosmetic satisfaction than TMLUR. The mean mid-term follow-up time for RALUR and TMLUR was 18.29 months and 24.64 months, respectively. Mid-term follow-up data showed that DUS was relieved with improved renal function after surgery in both groups, with no significant difference. Conclusions: RALUR and TMLUR are both safe and efficient for DUS treatment and achieve comparable mid-term outcomes in children. RALUR can reduce operative time and operative blood loss benefiting from its prominent technical superiority, but may currently bring about greater financial burden, with cosmetic satisfaction remaining to be improved.

## 1. Introduction

Benign distal ureteral strictures (DUS) in pediatrics are usually caused by urolithiasis, iatrogenic injury, infection, or simply because of congenital developmental abnormalities [1,2,3]. Ureteral obstruction due to stricture formation can lead to significant morbidity, such as chronic pain, infection, hydronephrosis and eventual irreversible renal injury, while posing a challenging disease entity for urologists due to the wide spectrum of treatment modalities required [4].

Ureteral reimplantation (UR) represents an established surgical treatment for DUS. The procedure might include a psoas hitch (PH) or a Boari flap (BF) to facilitate the creation of a tension-free anastomosis [5]. Historically, open UR (OUR) has offered high success rates of over 95% for DUS [6], while laparoscopic ureteral reimplantation (LUR), first reported by Winfield et al. in 1991 [7], has become a reasonable alternative to OUR, which has been proven to be a safe and feasible procedure and associated with lower intraoperative blood loss, lower operative pain and faster recovery [6,8]. By offering specialized technical advantages such as high-resolution three-dimensional view, tremor filtration with motion scaling and highly dexterous wrist-like instruments [4,9,10], the robotic surgical system dramatically improves laparoscopy by providing finer movement and easier intracorporeal suturing. Thus, robot assisted LUR (RALUR) is gradually substituting for conventional LUR as the first option for UR and achieved satisfactory results during the last decade in many high-volume centers [11,12].

In our center, to pursue better cosmetic results and shorter convalescence, the improved LUR system, trans-umbilical multiport laparoscopic ureteral reimplantation (TMLUR), was attempted to treat DUS. Meanwhile, since 2019, we have successfully applied RALUR for pediatric DUR. Up to date, there is a good amount of literature comparing the difference between RALUR and conventional LUR or OUR. However, TMLUR has not been discussed in reports and the differences between RALUR and TMLUR have not been analyzed. Therefore, based on the mid-term follow-up data from our center, this study aims to present the experience of TMLUR and assess the difference between RALUR and TMLUR.

## 2. Patients and Methods

### 2.1. Patient Selection

The cohorts of this study are patients aged less than 16 years old, diagnosed as DUS based on clinical symptoms, medical examination, laboratory tests and imagological diagnosis [13] and receiving RALUR or TMLUR at our center between April 2017 to November 2020. Selection of surgical protocols was based on the preference of patients and their families after informing of the potential benefits and limitations of both techniques preoperatively. Patients with previous or concomitant intra-abdominal surgery, complicated by vesicoureteral reflux or neurogenic bladder, or with bilateral intravesical ureteral reimplant were excluded. Through different entrances into the abdominal cavity, all these surgeries were performed by the same surgeon (Tong Q) skilled in laparoscopic techniques using routine laparoscopic instruments. All surgeries were performed by a transperitoneal approach.

Preoperative laboratory evaluation included serum creatinine, hemoglobin levels and urine culture. All patients routinely had cross-sectional imaging in the form of magnetic resonance urography (MRU), renal ultrasonography (USG) and renal scintigraphy as part of their initial diagnostic studies. For patients with nephrostomy tubes, antegrade re-photography was performed necessarily. Indications for surgery were repeated clinical symptoms, progressive hydronephrosis and ureteral dilatation, worsening renal function and clearly presented ureteral obstruction. Before discharging from hospital, patients routinely received renal USG. During follow-up, renal USG, renal scintigraphy and MRU were also usually performed to assess renal function before removing the double-J stent at 2 months after surgery, while another urinalysis or urine culture, urinary ultrasound and renal scintigraphy were repeated at 6–12 months and yearly thereafter.

### 2.2. Data Collection

Demographic and clinical characteristics of all patients were retrospectively collected from medical records and subsequent visits or contact in our hospital, comprising gender, age at the time of surgery, body mass index (BMI), DUS characteristics, abdominal surgical history, laterality, total length of incision, additional ports, operative time (including the docking time of RALUR), intraoperative blood loss, blood transfusion, length of stay, Vas pain score, complications (including wound infection, hematuria, UVJ leakage, subcutaneous emphysema, internal organ damage, stent migration and anastomotic stenosis), parents’ satisfaction scores (assessed by Client Satisfaction Questionnaire) [14] and follow-up data.

### 2.3. Ethical Statement

The present study was approved by the Institutional Ethics Committee of Union Hospital, Tongji Medical College, China. Informed consent was obtained from all parents of individual participants enrolled in the study.

### 2.4. Surgical Techniques

**RALUR:** Our RALUR technique is identical to that previously been reported in the literature [10], using the same basic principles applied for open extravesical ureter reimplantation. For a better intraperitoneal perspective, a trans-peritoneal approach was used to access the ureter and bladder. The patient was placed in the Trendelenburg position, with legs abducted on a split leg table. Then, the patient was secured to the operating table. A Foley catheter and nasogastric tube were routinely placed before surgery. A 12 mm port was inserted just into the umbilicus for the robotic camera port. On the midclavicular line of the affected side approximately 2–5 cm superior to the umbilicus, one 8-mm working robotic trocar was usually placed, while the other 8-mm working robotic trocar was positioned 2–5 cm inferior to the umbilicus at the contralateral midclavicular line. An assistant trocar (5-mm) was positioned between the camera and the 8-mm working robotic trocar superior to the umbilicus. The da Vinci Surgical System was docked in standard fashion (Figure 1A).

The surgical steps were almost identical to conventional LUR. First, the identification of the ureters was through the distal to the round ligament in females and vas deferens in males and their insertion in the bladder was noted (Figure 2A). The ureter was dissected distally to the point where stenosis existed and maintained periureteral blood supply (Figure 2B). A vessel loop placed around the ureter was conducive to atraumatic ureter handling, which was not always essential. Then, the ureter was transected near the area of the stenosis, cut up and the ureteral stump near the bladder was closed (Figure 2C). With the bladder partially filled, the detrusor muscle lengthwise was incised and separated for approximately 3 cm to expose the vesical mucosa by using sharp dissection on the lateral wall of the bottom of the bladder, leaving the mucosa intact (Figure 2D). A detrusor incision to ureteral diameter ratio of 4 or 5:1 was maintained. After the posterior anastomoses of the distal spatulated end of the ureter to the bladder was finished by 3-0 or 4-0 polyglactin sutures, a suitable multi-coil Double-J stent was placed in the ureter and bladder by the bedside assistant through the 5-mm port (Figure 2E). Then, the anterior anastomosis was completed. Finally, the detrusor muscle was closed by a 2-0 interrupted suture for an anti-reflux tunnel (Figure 2F). When the detrusor closing was completed, the bladder was filled with approximately 250 mL of saline to evaluate leakage and tension (Appendix A).

**TMLUR**: All patients were subjected to routine preoperative preparation and standard anesthesia protocol. The patients were similarly positioned in the Trendelenburg position. The 5-mm umbilical port for the camera and two 3-mm working ports were placed for TMLUR. The 5-mm port was placed at the umbilicus and secured with a skin suture. Notably, the other two 3-mm working trocars were laterally placed along subcutaneous planes before penetrating the peritoneum in TMLUR approach, established around the umbilicus, which might lead to more freedom of movement of the two instruments (Figure 1B). The procedures of TMLUR and RALUR in the abdominal cavity were similar. A vessel loop placed around the ureter was usually necessary. A “hitch stitch” of the bladder would make the incision of the detrusor muscle and the exposure of vesical mucosa easier. Specific steps were identical to the RALUR.

### 2.5. Statistical Analysis

Statistical analysis was performed using SPSS version 22.0 (IBM corporation, Armonk, NY, USA). Continuous variables were articulated as a mean ± standard deviation (SD), conforming to a normal distribution or presenting as median with range, were analyzed using Student *t*-test or Mann-Whitney U test. Categorical variables were presented as counts analyzed using the Chi-square test and Fisher’s exact test. *p* < 0.05 was considered statistically significant.

## 3. Results

### 3.1. Patient Demographics and Clinical Features

Patient demographics and clinical features were summarized in Table 1. The cohorts of RAMUR group (n = 28) and TMLUR group (n = 28) were comparable. Twenty-eight patients (20 boys and 8 girls), aged 57.48 months (ranged from 12.00 to 142.00 months), underwent RALUR with a mean BMI of 23.22 ± 1.84. In TMLUR group, there were 28 patients (18 boys and 10 girls), aged 52.62 months (ranged from 10.00 to 115.50 months), with a mean BMI of 22.75 ± 1.50. There were no significant differences between two groups in age (*p* = 0.522), gender (*p* = 0.567), laterality (*p* = 0.584), BMI (*p* = 0.298), clinical presentations (*p* = 0.577) and mean length of strictures (*p* = 0.720). The most frequented etiology of DUS was congenital stenosis in both groups. All patients were diagnosed as impaired renal function by ^99m^Tc-DTPA renal scan, based om the results of prolonged half maximum time (HMT) and decreased estimated glomerular filtration rate (eGFR) of affected kidney and there was no difference in renal function between two groups.

### 3.2. Comparison of Intra-Operative Characteristics in RALUR and TMLUR Groups

The detailed operative profiles were presented in Table 2. Mean operative time was significantly shorter for RALUR (127.11 ± 13.57 min) than TMLUR (138.39 ± 14.99 min, *p* = 0.005). Notably, the operative time of RALUR included surgery time and docking time. All surgeries in both groups were successfully performed without conversion to open surgery. There was a significantly lower hemorrhage volume in the RALUR group (18.57 ± 6.06 mL) compared with the TMLUR group (26.07 ± 9.36 mL, *p* = 0.001). The double-J stent was placed in each patient in both groups. No intraoperative complications and blood transfusion occurred in either group. Similar outcomes between RALUR and TMLUR group were noticed in oral feeding time (34.61 ± 6.01 vs. 35.82 ± 5.36 h, *p* = 0.428), length of hospital stay (6.07 ± 1.18 vs. 6.71 ± 1.49 d, *p* = 0.079), vas pain score at postoperative day 1 (3.50 ± 1.19 vs. 3.18 ± 1.06, *p* = 0.289), parents satisfaction score (30.07 ± 1.70 vs. 29.54 ± 2.13, *p* = 0.303) and catheter removal after surgery (5.14 ± 0.93 vs. 5.36 ± 1.06 d, *p* = 0.426).

### 3.3. Comparison of Post-Operative Complications in RALUR and TMLUR Groups

Before discharging from hospital, no patient suffered from wound infection, hernia formation, internal organ damage, or stent migration. One case in TMLUR developed urinary leakage which was cured by conservative therapeutics, while no urinary leakage was observed in RALUR group. Subcutaneous emphysema occurred in four cases in RALUR group and two patients in TMLUR group, respectively. (*p* = 0.388), while all were managed by compressing the emphysema area of the abdominal wall. Less economic pressures (23,265.45 ± 2673.99 RMB vs. 43,108.44 ± 2753.24 RMB; *p* < 0.001) were presented in the TMLUR group than the RALUR group (Table 2). Meanwhile, the incision for the RALUR group (approximate 33 mm in length) was larger and dispersed on the abdominal wall, when compared with total 11 mm incision centralized in the umbilicus of the TMLUR group (Figure 3).

### 3.4. Comparison of Follow-Up Data in RALUR and TMLUR Groups

Follow-up data and changes in affected renal function were presented in Table 3. The follow-up time was 18.29 ± 4.77 and 24.64 ± 4.65 months in RALUR and TMLUR groups, respectively (*p* < 0.001). In a six-month follow-up, all preoperative symptoms were resolved after surgery in RALUR group, but three patients in TMLUR group still had a little discomfort (two fever and one in abdominal pain) (*p* = 0.075). The three patients in TMLUR group were later diagnosed with anastomotic stenosis. One patient encountered anastomotic stenosis during the follow-up in RALUR group (*p* = 0.229). Among the four patients with anastomotic stenosis, two in TMLUR group and one in RALUR group were completely cured by antibiotics administration combined with the replacement of a D-J stent for another 2 months and the one in TMLUR group underwent re-operative RALUR afterward. They all needed another follow-up for 15-18 months for clinically free of obstruction while a longer follow-up was still needed to exclude the possibility of chronic obstruction or delayed recurrence. There were three patients emerging vesicoureteral reflux (VUR, two patients in Grade I and one patient in Grade II) in RALUR group and two patients (all in Grade II) in TMLUR group in six-month follow-up, whereas all these five patients did not suffer from any discomfort. One patient in RALUR group and one patient in TMLUR group got better on their own during the subsequent follow-up period. Follow-up evaluation included urinary routine, renal USG and renal scintigraphy. Considering the advantage of no radiation damage for children and the ability of observing blood supply, renal USG was generally used to evaluate the grade of VUR instead of VCUG in our center [15]. A distinct and similar improvement of renal function in the affected kidney compared with pre-operation was observed in both groups. (Table 3) The success rate was defined as freedom from radiographic or clinical evidence of ureterostenosis. The success rate of the RALUR and TMLUR groups were 100.00% and 96.43%, respectively (*p* = 0.313).

## 4. Discussion

Due to validation in earlier recovery, shorter hospital stays and better cosmetic results, conventional LUR is gradually becoming an alternate procedure of OUR for the treatment of DUS in children [6,8,16]. Although LUR was a minimally invasive surgical technique and can achieve a good cosmetic appearance, laparoendoscopic single-site surgery (LESS) decreased incisions and present better cosmetic outcomes. Nevertheless, the use of LESS in DUS was constrained by the requirement of specific multichannel laparoscopic ports and articulating or pre-bent instruments, especially in developing countries [17,18,19]. To achieve the identical invasiveness and cosmetic outcomes of LESS, after accumulating enough laparoscopic experience [20,21,22], we have tried and successfully performed many TMLUR surgeries by using the usual instruments in children, which has not been previously reported in the published literature. Recently, numbers of RALUR have increased annually and progressively overshadow conventional LUR due to its instinct technical advantages [4,9,10,11,12,23]. Meanwhile, since urologists in our center are proficient in robot-assisted laparoscopic skills, since 2019 RALUR is applied for DUS in pediatric patients. Nevertheless, limited to the advanced skills of TMLUR and the exorbitant platform of RALUR, only a small quantity of urologists can simultaneously repair DUS with these two procedures and evaluate their distinction. Hence, this study presented the first nationwide sample comparing RALUR and TMLUR in the pediatric population and investigates the merits and demerits of TMLUR and the difference between TMLUR and RALUR.

In this study, we presented a retrospective comparative study of 56 patients, including 28 TMLUR and 28 RALUR. The function of the affected side of the kidney was obviously impaired based on the results of prolonged HMT and decreased eGFR measured by ^99m^Tc-DTPA renal scan, while the other kidney currently still met the needs of the body as indicated by normal serum creatinine levels. Notably, no difference in oral feeding time, length of stay, vas pain score at postoperative day 1, patient satisfaction score, or catheter removal was noticed between the two groups. Meanwhile, there were no patients converting to OUR during the surgery. No differences of surgical success rate, perioperative complications, or follow-up results were found in this study, which indicate that TMLUR is a safe procedure for the management of DUS in children with comparable efficiencies to RALUR.

From a technical point of view, TMLUR, an improvement on LUR, is a challenging procedure to perform and requires strong laparoscopic experience and advanced laparoscopic skills including intracorporeal suturing and knotting. Furthermore, it is difficult ergonomically for the surgeon and has not reached a widespread diffusion among pediatric urologists, due to high technical difficulty and a long learning curve [24]. However, RALUR mitigates these shortcomings on account of articulating robotic wrists, enhanced manipulation and visualization. In this study, significantly less blood loss was observed in the RALUR group, which means a lower risk of vascular injury or inadvertent coagulation and avoids the possibility of insufficient blood supply. The blood supply of the ureter affects ureteral viability severely, which might lead to anastomotic stenosis and urinary leakage. Meanwhile, operative time of RALUR was shorter than TMLUR in our cohorts, even including docking time, reaching a consensus with previous studies [9,23]. All these certify to the technical strength and convenience of RALUR in intracorporal elaborate operation. Compared with the long learning curve of TMLUR, there seems to be a definite learning curve for RALUR. Gundeti et al. [25] and Sachdev et al. [26] listed several modifications incorporated along their RALUR learning curve, which seemed to improve their success and has helped reduce complications over the years. We believe the steeper learning curve and simplified tasks will ultimately make RALUR a minimally invasive approach performed by a larger number of surgeons, so that even relatively inexperienced surgeons might be attempting RALUR.

Nevertheless, in other respects, TMLUR has certain advantages over RALUR for DUS in children. Primarily, robotic surgery has been criticized for the prohibitive cost of the platform and disposable instruments [27], which may increase expenditure for patients. Consistently, we found higher hospitalization expenses of the RALUR group in this study. Similarly, Bowen et al. reported that operating room charges were higher for the robotic cohort, with shorter length of hospital stay, when compared with OUR in children [6]. We believe that patients will further benefit from RALUR procedure with the development of low-cost robotic system in the future. In addition, although both RALUR and TMLUR are minimally invasive surgical options to correct DUS in children, a better cosmetic postoperative appearance is achieved in TMLUR with total 11 mm incision centralized in the umbilicus. When considering the economic or cosmetic effectiveness in the first place, TMULR might be the preferred choice in terms of current technology.

## 5. Conclusions

In summary, our findings indicate that RALUR and TMLUR are both safe and efficient for DUS treatment and achieve comparable mid-term outcomes in children. RALUR can reduce operative time and operative blood loss benefited from its prominent technical superiority, which may currently bring about a greater financial burden, with cosmetic satisfaction remaining to be improved.

## 6. Limitations

This study was a retrospective study, comprising many other confounding factors, including age, weight, sex and so on, which might influence the therapeutic effect, rather than the different surgical methods. Meanwhile, a relatively small number of patients were involved and the follow-up period was not sufficient. These may limit the statistical analysis of the results and reporting conclusions.

## Figures and Tables

**Figure 1 jcm-11-06229-f001:**
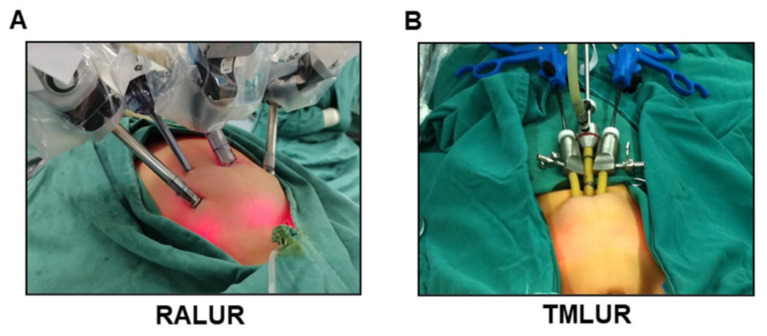
Patient position in RMLUR and TMLUR procedures. (**A**) Patient and trocars’ positioning in RMLUR. (**B**) Patient and trocars’ positioning in TMLUR.

**Figure 2 jcm-11-06229-f002:**
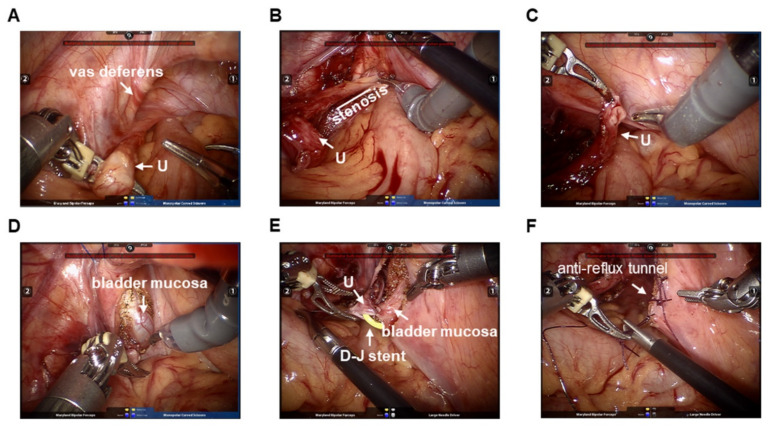
Surgical steps of RMLUR for DUS in children. (**A**) Insertion of ureter in bladder. (**B**) Ureter is dissected distally to the point where stenosis exists and maintained periureteral blood supply. (**C**) Ureter is transected near the area of the stenosis and cut up. (**D**) Incising and separating the detrusor muscle lengthwise for approximately 3 cm to expose the vesical mucosa. (**E**) Placing multi-coil Double-J stent in the ureter and bladder. (**F**) Detrusor muscle is closed by a 2-0 interrupted suture for an anti-reflux tunnel.

**Figure 3 jcm-11-06229-f003:**
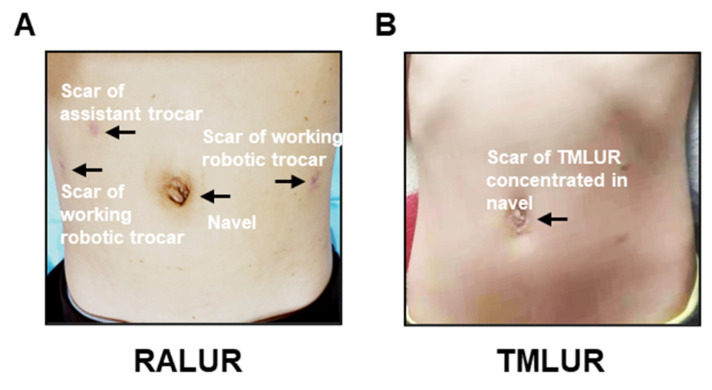
Postoperative scar in abdominal wall. (**A**) Postoperative scar in abdominal wall of RMLUR. (**B**) Postoperative scar in abdominal wall of TMLUR.

**Table 1 jcm-11-06229-t001:** Patient demographics and clinical features in RALUR and TMLUR groups.

	RALUR	TMLUR	*p* Value
Case number	28	28	NA
Age (months)	57.48 ± 30.66	52.62 ± 25.52	0.522
	(Range 12.0–142.0)	(Range 10.0–115.5)	
Gender (F/M)	20/8	18/10	0.567
BMI	23.22 ± 1.84	22.75 ± 1.50	0.298
Laterality			0.584
Left	16 (57.1%)	18 (64.3%)	
Right	12 (42.9%)	10 (35.7%)	
Etiology			0.495
Congenital	21 (75.0%)	23 (82.1%)	
Urolithiasis	5 (17.9%)	3 (10.7%)	
Infection	2 (7.1%)	2 (7.1%)	
Length (cm)	0.64 ± 0.15	0.60 ± 0.22	0.720
Clinical presentation			0.577
Yes	19 (67.9%)	17 (60.7%)	
Fever	6	6	
Flank pain	12	10	
UTI	14	15	
No	9 (32.1%)	11 (39.3%)	
Urinalysis/urine culture	18 (64.3%)	19 (67.9%)	0.778
Ultrasonography	30.10 ± 6.55	29.11 ± 7.44	0.600
99mTc-DTPA renal scan of affected side			0.501
HMT > 20 min	18 (64.3%)	17 (60.7%)	
HMT 15–20 min	10 (35.7%)	11 (39.3%)	
eGFR of affected side	58.11 ± 9.77	59.36 ± 11.11	0.657
SCr (μmol/L)	43.71 ± 10.46	44.13 ± 9.64	0.478

RALUR, robot-assisted laparoscopic ureteral reimplantation; TMLUR, trans-umbilical multiport laparoscopic ureteral reimplantation; UTI, urinary tract infections; HMT, half maximum time; eGFR, estimated glomerular filtration rate; SCr, serum creatinine.

**Table 2 jcm-11-06229-t002:** Intraoperative and postoperative outcomes in RALUR and TMLUR groups.

	RALUR	TMLUR	*p* Value
Case number	28	28	NA
Operative time (minutes)	127.11 ± 13.57	138.39 ± 14.99	0.005
Estimated blood loss (ml)	18.57 ± 6.06	26.07 ± 9.36	0.001
D-J placement	28	28	NA
Conversion to open	0	0	NA
Oral feeding time (hours)	34.61 ± 6.01	35.82 ± 5.36	0.428
Length of stay (days)	6.07 ± 1.18	6.71 ± 1.49	0.079
Vas pain score at postoperative day 1	3.50 ± 1.19	3.18 ± 1.06	0.289
Parents satisfaction score	30.07 ± 1.70	29.54 ± 2.13	0.303
Catheter removal (days)	5.14 ± 0.93	5.36 ± 1.06	0.426
Postoperative Complication			
Wound infection	0	0	NA
Hernia formation	0	0	NA
Urinary leakage	0	1 (3.6%)	0.313
Subcutaneous emphysema	4 (14.3%)	2 (7.1%)	0.388
Internal organ damage	0	0	NA
Stent migration	0	0	NA
Hospitalization expenses	43108.44 ± 2753.24	23265.45 ± 2673.99	<0.001

RALUR, robot-assisted laparoscopic ureteral reimplantation; TMLUR, trans-umbilical multiport laparoscopic ureteral reimplantation.

**Table 3 jcm-11-06229-t003:** Follow-up data in RALUR and TMLUR groups.

	RALUR	TMLUR	*p* Value
Case number	28	28	NA
Follow up in months	18.29 ± 4.77	24.64 ± 4.65	<0.001
Symptom remissions	18 (100%)	14 (82.4%)	0.075
Urinalysis/urine culture	2 (7.1%) *	5 (17.9%) ^#^	0.225
Ultrasonography	13.77 ± 5.17 *	14.33 ± 5.81 ^#^	0.704
99mTc-DTPA renal scan of affected side			0.690
HMT > 20 min	0 *	1 (3.6%) ^#^	
HMT 15–20 min	3 (10.7%) *	1 (3.6%) ^#^	
HMT < 15 min	25 (89.3%) *	26 (92.8%) ^#^	
eGFR of affected side at 12 months postoperatively	86.05 ± 10.99 *	85.80 ± 12.32 ^#^	0.936
SCr (μmol/L)	44.09 ± 11.26	46.05 ± 10.88	0.307
Anastomotic stenosis	1 (3.6%)	3 (10.7%)	0.299
VUR			0.167
GradeⅠ	2 (7.2%)	0	
GradeⅡ	1 (3.6%)	2 (7.2%)	
Reoperation	0	1 (3.6%)	0.313
Success rate	28 (100.0%)	27 (96.4%)	0.313

RALUR, robot-assisted laparoscopic ureteral reimplantation; TMLUR, trans-umbilical multiport laparoscopic ureteral reimplantation; HMT, half maximum time; eGFR, estimated glomerular filtration rate; SCr, serum creatinine; VUR, vesicoureteric reflux. * *p* < 0.05 vs. preoperative data in RALUR group; ^#^
*p* < 0.05 vs. preoperative data in TMLUR group.

## Data Availability

All the obtained data used to support the findings of this study are available from the corresponding author upon reasonable request.

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
