# Peer review of "Robot-Assisted versus Trans-Umbilical Multiport Laparoscopic Ureteral Reimplantation for Pediatric Benign Distal Ureteral Stricture: Mid-Term Results at a Single Center"

_jcm, 2022, doi:10.3390/jcm11216229_

Round 1

Reviewer 1 Report

Dear Authors,

I read the paper “Robot-assisted versus trans-umbilical multiport laparoscopic ureteral reimplantation for pediatric benign distal ureteral stricture: Mid-term results at a single center” with interest.

The aim of the study was to compare robot-assisted vs tranumbilical laparoscopic approach for ureteral reimplantation. Despite some surgical advantages (operative time and blood loss), no functional differences were found between the two groups.

Overall the paper is interesting. Few remarks:

Abstract: you mentioned this was a matched paired comparison. Instead this is a retrospective evaluation of two groups without a matched paired analysis. Please correct.

Title: adequate

Introduction: adequate.

Methods: adequate.

-       Concerning the surgical technique you described in this section, Mai I suggest to add a brief video as supplementary material? Just few minutes with the most important step of the intervention may increase surgical interest of your work.

Results:

-       Could you specify the cause and the mean length of strictures of you patients? Was there any difference between the two groups?

-       Could you provide also the serum creatinine levels and renal scan data please? You should then compare the pre- and postoperative data between the two groups

-        

Discussion:

-       Please add a brief section summarizing the main limitations of your work (including retrospective study design, low sample size, short follow-up)

Finally, I believe the paper needs a language revision.

Author Response

Reviewer #1:

  1. Abstract: you mentioned this was a matched paired comparison. Instead this is a retrospective evaluation of two groups without a matched paired analysis. Please correct.

Responses: We appreciate the reviewer’s positive comments and revision guidance on our manuscript. We are sorry for incorrect description as “matched paired comparison” in article, and we have corrected the “matched paired comparison” to “retrospective comparison” in this revised manuscript.

  1. Concerning the surgical technique you described in this section, Mai I suggest to add a brief video as supplementary material? Just few minutes with the most important step of the intervention may increase surgical interest of your work.

Responses: Good comments. A brief surgical video comprising critical steps was provided as a supplementary material, including dissection of distal ureter, transection of ureter near the stenosis, filling of bladder, incision of muscular layer, exposure of vesical mucosa, anastomoses of ureter to bladder, placement of double-J stent, and closure of detrusor muscle. We believe that it is helpful for better understanding the surgical procedure in this way.

  1. Could you specify the cause and the mean length of strictures of you patients? Was there any difference between the two groups?

Responses: The cause and mean length of strictures were added into Table 1. The most frequented etiology of benign distal ureteral stricture (DUS) was congenital stenosis in both groups (21 in RALUR group and 23 in TMLUR group). The mean length of strictures was provided in RALUR (0.64 ± 0.15 cm) and TMLUR (0.60 ± 0.22 cm) group. There was no significant difference of these two factors between two groups (P=0.495 for etiology and P=0.720 for length of strictures).

  1. Could you provide also the serum creatinine levels and renal scan data please? You should then compare the pre- and postoperative data between the two groups.

Responses: Good comments. Dynamic radionuclide renography is able to provide visual images and time-activity curves, and serves as a common and effective method for evaluating renal function (Nucl Med Commun, 2001, 9: 987-95). As a reliable diagnostic criteria, ureteral stricture is indicated when half maximum time (HMT) is longer than 20 minutes. In our work, 99mTc-DTPA renal scan and serum creatinine measurement were routinely performed before and after surgery. As required, we have provided the results of half maximum time (HMT) and estimated glomerular filtration rate (eGFR) of affected kidney into Table 1 and Table 3, respectively. Our results showed no significant difference in HMT and eGFR between RALUR and TMLUR groups before the surgery, while serum creatinine levels also maintained at a normal level, due to reserved function of another side kidney. After the surgery, both RALUR and TMLUR improved the impaired split renal function efficiently, without significant difference.

  1. Please add a brief section summarizing the main limitations of your work (including retrospective study design, low sample size, short follow-up).

Responses: In this revised manuscript, we have addressed the limitations of this study, such as limited patients and insufficient follow-up time, which warrant further investigation. 

Reviewer 2 Report

This study is a retrospective non randomized study about ureteral reimplantation for pediatric benign stricture by robotics or laparoscopy.

Twenty eight patients were included in each group.

The paper needs to be reviewed by a native English person as mistakes are common.

The abstract is informative.

In material and methods, the surgical indication should be more precise. In the follow-up, to do MRU yearly is non sense if the patient is asymptomatic with a normal US. On the contrary, they do not perform MCUG to exclude VUR. Why?

The techniques are well explained with nice pictures.

In the results, we should have the type and etiology of stenosis (congenital, other). 

The symptoms should be described: febrile UTIs, pain, stone, …

In the table, eGFR is non sense as the other kidney is likely normal. It should be differential function on nuclear medicine study. 

For operative time, in the methods, it is from skin to skin. In the results, it includes docking for the robotic group which is logical. This needs to be corrected. The volume of bleeding has no sense as for this type of surgery, it is negligible.

Oral feeding, hospital stay as very long for the 2 cohorts.

The patients satisfactory score should be replaced by parents satisfaction score due to the age of the patients.

The discussion and references are quite good. The reference 2 is about urolithiasis which is not the topic of this study.

Author Response

Reviewer #2:

  1. In material and methods, the surgical indication should be more precise. In the follow-up, to do MRU yearly is non sense if the patient is asymptomatic with a normal US. On the contrary, they do not perform MCUG to exclude VUR. Why?

Responses: Good comments. The indication for surgery were repeated clinical symptoms, progressive hydronephrosis and ureteral dilatation, worsening renal function, and clearly presented ureteral obstruction. We are sorry for unclear description of follow-up evaluation. MRU was performed before removing the double-J stent at 2 months after surgery and in patients who suffered from recurrent stenosis, while urinary routine, renal ultrasonography (USG), and renal scintigraphy were conventionally performed during follow-up. Considering the advantage of no radiation damage for children and the ability of observing blood supply (Iran J Kidney Dis, 2021, 5: 328-335), renal USG was generally used to evaluate the grade of VUR instead of VCUG in our center. There were three patients emerging vesicoureteral reflux (VUR, two patients in Grade I and one patient in Grade II) in RALUR group, and two patients (all in Grade II) in TMLUR group in six-month follow-up. All these five patients did not suffer from any discomfort. One patient in RALUR group and one patient in TMLUR group got better on their own during subsequent follow-up period. And other patients took periodic outpatient follow-up. In this revised manuscript, we have addressed these clearly as required.

  1. In the results, we should have the type and etiology of stenosis (congenital, other) and the symptoms should be described: febrile UTIs, pain, stone.

Responses: Good comments. The etiology and specific symptoms of patients were added in Table 1. The most frequented etiology was congenital stenosis, while the most frequented clinical presentation was urinary tract infections.

  1. In the table, eGFR is non sense as the other kidney is likely normal. It should be differential function on nuclear medicine study.

Responses: Good comments. The presented eGFR was the result of affected kidney measured by 99mTc-DTPA renal scan. To avoid confusion, we have corrected this description in revised manuscript and Tables.

  1. For operative time, in the methods, it is from skin to skin. In the results, it includes docking for the robotic group which is logical. This needs to be corrected. The volume of bleeding has no sense as for this type of surgery, it is negligible.

Responses: Good comments. The surgery time in methods was corrected. When dissociating the ureter and incising the muscular layer of bladder, it was inevitable to damage some small blood vessels. Due to easier manipulation and improved visualization, the blood loss was significantly less in RALUR group, than that of TMLUR group, which may indicate lower possibility of vascular injury or inadvertent coagulation in robotic surgery, which warrant further studies.

  1. Oral feeding, hospital stay as very long for the 2 cohorts.

Responses: On account of underdeveloped community hospitals in our country, patients generally discharge from hospital until they are almost recovered after surgery. That was why the time of oral feeding and hospital stay were longer. As a result, similar length of hospital stay is reported by other medical center in our country (Urol J, 2020, 173: 252-256).

  1. The patients satisfactory score should be replaced by parents satisfaction score due to the age of the patients and the reference 2 is about urolithiasis which is not the topic of this study.

Responses: In this revised manuscript, the inappropriate description of “patients satisfactory score” and citation of second reference were corrected.

Round 2

Reviewer 1 Report

Dear Authors, I think that paper has been improved significantly.

Reviewer 2 Report

The paper is significantly improved due to the modifications after reviw